# Receptor Tyrosine Kinases Amplified in Diffuse-Type Gastric Carcinoma: Potential Targeted Therapies and Novel Downstream Effectors

**DOI:** 10.3390/cancers14153750

**Published:** 2022-08-01

**Authors:** Hideki Yamaguchi, Yuko Nagamura, Makoto Miyazaki

**Affiliations:** Department of Cancer Cell Research, Sasaki Institute, Sasaki Foundation, Tokyo 101-0062, Japan; nagamura@po.kyoundo.jp (Y.N.); m-miyazaki@po.kyoundo.jp (M.M.)

**Keywords:** diffuse-type gastric carcinoma, gene amplification, peritoneal dissemination, receptor tyrosine kinase

## Abstract

**Simple Summary:**

Diffuse-type gastric carcinoma (DGC) is an aggressive subtype of gastric carcinoma with an extremely poor prognosis due to frequent peritoneal metastasis and high probability of recurrence. Its pathogenesis is poorly understood, and consequently, no effective molecular targeted therapy is available. The importance of oncogenic receptor tyrosine kinase (RTK) signaling has been recently demonstrated in the malignant progression of DGC. In particular, RTK gene amplification appears to accelerate peritoneal metastasis. In this review, we provide an overview of RTK gene amplification in DGC and the potential of related targeted therapies.

**Abstract:**

Gastric cancer (GC) is a major cause of cancer-related death worldwide. Patients with an aggressive subtype of GC, known as diffuse-type gastric carcinoma (DGC), have extremely poor prognoses. DGC is characterized by rapid infiltrative growth, massive desmoplastic stroma, frequent peritoneal metastasis, and high probability of recurrence. These clinical features and progression patterns of DGC substantially differ from those of other GC subtypes, suggesting the existence of specific oncogenic signals. The importance of gene amplification and the resulting aberrant activation of receptor tyrosine kinase (RTK) signaling in the malignant progression of DGC is becoming apparent. Here, we review the characteristics of RTK gene amplification in DGC and its importance in peritoneal metastasis. These insights may potentially lead to new targeted therapeutics.

## 1. Introduction

Gastric cancer (GC) has the fifth-highest incidence among cancers, and it is the fourth leading cause of cancer-related death worldwide [1]. According to Lauren’s classification, GC is histologically categorized into two main subtypes: intestinal-type gastric carcinoma (IGC) and diffuse-type gastric carcinoma (DGC) [2]. DGC is an aggressive subtype with an extremely poor prognosis owing to rapid infiltrative invasion within the submucosa and frequent occurrence of peritoneal dissemination and high probability of recurrence [3]. DGC contains poorly differentiated and signet ring carcinoma cells that solitarily exist within a dense tumor stroma because of the lack of cell–cell adhesion. Scirrhous gastric carcinoma (SGC), also referred to as linitis plastica, is a subtype of DGC that is characterized by an extensive desmoplastic reaction [4].

Peritoneal metastasis is the dissemination of cancer to the peritoneum, which covers the abdominal cavity and the intra-abdominal organs, and it is often associated with the formation of malignant ascites. The quality of life and survival rate of the patients are both greatly reduced by peritoneal metastasis. However, no effective molecular targeted therapy is currently available. Because the pathological and histological features of DGC are quite different from those of IGC, studies focusing on DGC are necessary to understand its molecular basis and to develop effective molecular targeted therapy. The importance of oncogenic receptor tyrosine kinase (RTK) signaling has been recently demonstrated in the malignant progression of DGC. In particular, RTK gene amplification appears to accelerate peritoneal metastasis, and thus it is considered a promising therapeutic target. In this review, we provide an overview of RTK gene amplification in DGC and the potential of related targeted therapies. The recent literature regarding the downstream effectors of RTKs is summarized. Finally, we highlight issues that should be addressed to effectively target amplified RTK in DGC.

## 2. Gene Amplification of RTKs in DGC

RTKs are cell surface receptors that regulate various cellular processes, including proliferation, survival, metabolism, differentiation, migration, and invasion. Upon binding to specific ligands, two RTK molecules dimerize and tyrosine-phosphorylate each other, and recruit and activate a variety of intracellular signaling molecules to trigger the activation of downstream signaling pathways, such as Ras/MAPK, PI3-kinase/Akt, JAK/Stat, and NF-κB pathways (Figure 1). RTKs are well-known oncogenic drivers that are aberrantly activated in a wide range of human cancers. Oncogenic activation of RTK signaling is caused in several ways, including overexpression, activating mutations, fusions, and ligand dysregulation [5]. RTK overexpression is commonly caused by gene amplification that provokes ligand-independent dimerization owing to local enrichment, causing constitutive activation of downstream signaling pathways. Cancer cells harboring gene amplification of RTKs often exhibit ‘oncogene addiction’, and their growth and survival are highly dependent on the activity of amplified RTKs. Hence, RTKs amplified in cancers have been considered promising therapeutic targets; many RTK inhibitors have been developed, evaluated in preclinical and clinical trials, and used in the clinic.

The amplification of RTK genes has been identified in GC, including *EGFR* encoding epidermal growth factor receptor (EGFR), *ERBB2* encoding human epidermal growth factor receptor 2 (HER2, also known as ERBB2), *MET* encoding Met (also known as c-Met), and *FGFR2* encoding fibroblast growth factor receptor 2 (FGFR2), as summarized in Table 1. We also performed gene-amplification analysis of these RTKs in GC using cBioPortal and several publicly available datasets (Table 2). The frequencies of gene amplification were as follows: *EGFR*, 2.7–14.7%; *ERBB2*, 8–22.0%; *MET*, 2.2–8.8%; *FGFR2*, 2–5.4%. This is roughly consistent with other large cohort studies and confirms that a subset of GC harbors amplification of these RTK genes.

Tsujino et al. reported that amplification of *EGFR* and *ERBB2* occurs in poorly differentiated adenocarcinoma and signet ring cell carcinoma, i.e., DGC [6]. The incidence of *EGFR* amplification is higher in metastatic tumors than that in primary tumors. Moreover, EGFR overexpression is frequently observed in GC and is associated with worse prognosis (Table 1). A large cohort study reported that *ERBB2* is amplified in approximately 23% of GC cases [17]. The rate of *ERBB2* amplification was substantially higher in patients with IGC than that in those with DGC in several studies that analyzed different large cohorts (Table 1). Therefore, HER2 is less likely to play a specific role in DGC but is generally implicated in the progression of GC.

Met is a receptor for hepatocyte growth factor (HGF) and encoded by *MET* proto-oncogene [40]. Physiologically, Met plays an essential role in embryogenesis and tissue regeneration [41]. Met is aberrantly activated in various cancer types owing to gene amplification, point mutations, rearrangement, overexpression, and aberrant splicing [42]. Oncogenic activation of Met promotes diverse malignant aspects of tumors, including invasion, metastasis, drug resistance, and angiogenesis [41]. Kuniyasu et al. reported that *MET* amplification frequently occurs in SGC cell lines and tumor tissues [21]. Following studies also reported *MET* amplification in GC, although the rate differs among studies, and its association with DGC and unfavorable outcomes (Table 1).

FGFR2 is a receptor for fibroblast growth factors, and it is oncogenically activated by genetic alterations in cancer [43]. FGFR2 is identical to the product of the *K-sam* gene that was cloned as a gene amplified in the KATO-III SGC cell line [44]. FGFR2 overexpression caused by gene amplification is substantially more frequent in DGC and metastatic GCs than in primary GCs, and it is associated with tumor progression and poor patient survival [43] (Table 1).

The Cancer Genome Atlas (TCGA) project classified GC into four molecular subtypes: Epstein–Barr virus-positive, microsatellite-unstable, genomically stable, and chromosomally instable [10]. DGC is enriched in the genomically stable subtype and is characterized by recurrent E-cadherin and RhoA mutations or fusions of Rho GAPs. However, alterations in RTK genes are rare in the genomically stable subtype: *EGFR* (2%), *ERBB2* (7%), *FGFR2* (9%), and *MET* (0%). In contrast, a recent multiomics study demonstrated that alterations in RTK and downstream MAPK signaling pathways, mostly gene amplification of *KRAS* (19.4%), *FGFR2* (11.2%), *MET* (7.1%), *ERBB2* (5.1%), and *EGFR* (4.1%), occur more frequently in cancer cells within malignant ascites than those in primary DGC [12] (Figure 1). In addition, RTK alterations occur in a mutually exclusive manner [8,9,12]. In contrast, *MET* amplification has been reported to coexist with *HER2* amplification in several tumor samples [18]. Interestingly, Tajiri et al. reported that *HER2* was coamplified with *EGFR, FGFR*, and *MET* in some tumors, but in mutually exclusive cells [15]. Thus, the amplification of one RTK signaling component may be sufficient to drive tumor progression and can occur in a mutually exclusive manner at the cellular level. Taken together, RTK amplification is most likely a key event in the acquisition of malignant and metastatic phenotypes in DGC cells.

Loss of E-cadherin function that weakens cell–cell adhesion is a hallmark of DGC. Germline mutation of *CDH1* gene encoding E-cadherin causes hereditary DGC [45]. Previous multiomics studies did not show any correlation between *CDH1* mutation and RTK gene amplification [10,12]. Nevertheless, RTK signaling can negatively regulate E-cadherin function via downregulation/degradation of E-cadherin and disassembly of E-cadherin complexes [46]. Thus, RTK gene amplification and resulting oncogenic activation may contribute to the loss of E-cadherin function in DGC without *CDH1* mutation.

## 3. Targeting RTKs for Peritoneal Dissemination of DGC

In general, DGC is less sensitive to cytotoxic chemotherapy than IGC [3,47]. Although the anti-HER2 antibody, trastuzumab, in combination with chemotherapy improved the survival of patients with HER2-positive advanced IGC, its efficacy was limited in patients with DGC [48]. This is most likely because the rate of *HER2* amplification was low in DGC. In contrast, dozens of RTK inhibitors targeting MET, FGFR2, and EGF exhibited efficacy against DGC harboring RTK gene amplification, at least in vitro and in preclinical models (Table 3).

We and others have shown that Met inhibitors, including PHA-665752, capmatinib, crizotinib, and E7050, exhibit remarkable antitumor and/or antiperitoneal dissemination activities against DGC cells that are positive for *MET* amplification in mouse xenograft models [8,12,54,55,56]. Some Met inhibitors, such as crizotinib, AMG 337, and Savolitinib, showed antitumor activity in patients with *MET*-amplified GC in clinical trials [8,51,64]. The monoclonal antibody, ABT-700, which disrupts Met dimerization, also showed efficacy against *MET*-amplified DGC [18]. Moreover, P3D12-vc-MMAF, which is a conjugate of a Met-degrading antibody and the tubulin inhibitor, MMAF, exhibited a drastic antitumor efficacy in DGC with *MET* amplification [59].

Gene alterations in the FGFR family (FGFR1, FGFR2, FGFR3, and FGFR4) have been reported in various cancers. Thus, pan-FGFR inhibitors have been developed and show antitumor effects against DGC in preclinical studies, and some of them have been tested in clinical trials [43,84]. As gene amplification of FGFR2 is the predominant FGFR alteration in DGC, the development of FGFR2-specific inhibitors may improve efficacy and reduce adverse effects. In this regard, a unique selective allosteric inhibitor of FGFR2, alofanib (RPT835), was reported [85]. Although its efficacy against DGC is currently unclear, clinical studies on alofanib in patients with advanced or metastatic GC are ongoing. Other therapeutic modalities targeting FGFR2, such as a bivalent degrader, DGY-09-192 [78], a neutralizing monoclonal antibody, PRO-007 [80], and an antibody–drug conjugate, BAY 1187982 [81], may have better efficacy and selectivity than that of small-molecule inhibitors.

Although DGC cells addicted to amplified RTK signaling firstly showed high sensitivity to the RTK inhibitors, resistant cells emerge upon their continuous exposure to RTK inhibitors. As this is a serious issue in clinical usage of RTK inhibitors, the underlying molecular mechanisms have been extensively studied. Coamplification of RTKs such as HER2 and/or EGFR contribute to therapeutic resistance to Met inhibitor in DGC harboring Met amplification [86]. A patient with concurrent *MET* and *HER2* amplification responded to combined MET/HER2 inhibition. It was reported that FGFR2 overexpression is responsible for Met inhibitor resistance in *MET*-amplified PDX tumors of GC [28]. In this case, treatment with FGFR2 and Met inhibitors blocked the tumor growth. Overexpression of phosphoinositide 3-kinase (PI3K) p110α contributes to acquired resistance to Met inhibitor in DGC cells with *MET* amplification [83]. PI-103, a PI3K inhibitor, in combination with Met inhibitor showed antitumor effects against the Met-inhibitor-resistant DGC cells. Truncated forms of RAFs were also reported to confer resistance to Met inhibition in DGC cells with *MET* amplification [87]. FGFR2-ACSL5 fusion was found in a patient with *FGFR2*-amplified GC that acquired resistance to FGFR inhibitor [88]. Acquired resistance to FGFR inhibitor also occurs via PKC-mediated GSK3β phosphorylation in DGC-derived PDX tumors with *FGFR2* amplification [89]. PKC inhibitors reversed the resistance of FGFR2-addicted tumors to FGFR inhibitors in vivo. Futibatinib, a unique irreversible pan-FGFR inhibitor that binds to the FGFR kinase domain, demonstrates antitumor activity against DGC with *FGFR2* amplification [70,90]. Notably, futibatinib can inhibit FGFR2 mutants resistant to ATP-competitive FGFR inhibitors [70]. Taken together, the identification of responsible and druggable molecules and the development of inhibitors with different modes of action are critical to overcoming resistance to RTK inhibitors in RTK-addicted DGC.

RTK signaling plays a pivotal tumor-supporting role not only in cancer cells, but also in the tumor microenvironment. For example, vascular endothelial growth factor receptor (VEGFR) expressed in endothelial cells promotes tumor angiogenesis. Accordingly, dual- or multitarget tyrosine kinase inhibitors that inhibit both VEGFR in endothelial cells and other RTKs in cancer cells may have better therapeutic effects against RTK-addicted tumors. Several inhibitors targeting both VEGFR and Met or FGFR2 have been developed and tested in preclinical and clinical trials (Table 3). However, targeting multiple RTKs may increase the risk of adverse effects.

## 4. Novel Downstream Effectors of RTK in DGC

As described above, the use of RTK inhibitors inevitably results in acquired resistance in tumors. Therefore, understanding the downstream signaling of RTK is critical for the development of novel and alternative targeting approaches. PLEKHA5 (Pleckstrin homology domain containing A5) is a member of the PLEKHA family of proteins that contain a PH domain. Using a phosphoproteomic approach, we identified PLEKHA5 as a protein that is tyrosine-phosphorylated downstream of Met signaling (Figure 2) [91]. PLEKHA5 silencing selectively blocked the growth of DGC cells addicted to amplified Met, even when they acquired resistance to Met inhibitors. PLEKHA5 knockdown dysregulates glycolysis, leading to JNK activation and apoptotic cell death. In a mouse xenograft model, PLEKHA5 silencing markedly suppressed the peritoneal dissemination of *MET*-amplified DGC cells. Although its precise cellular functions remain to be elucidated, PLEKHA5 may be a biomarker for Met addiction, as well as a potential therapeutic target. 

SHP2 (Src homology region 2 domain-containing phosphatase 2, also known as PTPN11, protein-tyrosine phosphatase nonreceptor type 11) is an oncogenic nonreceptor-type tyrosine phosphatase that regulates ERK activation downstream of RTK signaling [92]. Recurrent mutations in the *PTPN11* gene encoding SHP2 have been observed in a variety of human cancers; therefore, SHP2 is thought to be a promising therapeutic target. Consequently, effective and selective allosteric SHP2 inhibitors, such as SHP099, have been developed recently [93]. We identified SHP2 by phosphoproteomic analysis of DGC cells with *MET* amplification [82] (Figure 2). SHP2 was tyrosine-phosphorylated in DGC cells with either *MET* or *FGFR2* gene amplification. The growth of these cells was severely impaired by the knockdown or pharmacological inhibition of SHP2, even in Met-addicted DGC cells that acquired resistance to Met inhibitors. Furthermore, inhibition of SHP2 markedly suppressed the peritoneal dissemination of DGC cells harboring *MET* amplification. As SHP2 serves as a common signaling node downstream of multiple RTKs, targeting SHP2 may be an attractive alternative approach for the treatment of DGC with RTK amplification. As SHP2 is also involved in the immune checkpoint downstream of PD-1 [94], blockage of SHP2 may show additional efficacy through dual inhibition of RTK signaling and immune suppression.

Transferrin receptor 1 (TfR1) is a ubiquitously expressed membrane protein necessary for the cellular uptake of iron-loaded transferrin. TfR1 is upregulated in a variety of cancers and supports cancer cell growth by fulfilling an increased iron demand [95]. Shirakihara et al. recently identified TfR1 as a tyrosine-phosphorylated protein associated with FGFR2 in DGC harboring *FGFR2* amplification [96] (Figure 2). TfR1 knockdown or FGFR2 inhibition impaired iron uptake and proliferation in DGC cells with *FGFR2* amplification. Furthermore, TfR1 knockdown suppressed peritoneal metastasis of *FGFR2*-amplified DGC and improved survival in a mouse xenograft model. Thus, TfR1 plays a pivotal role in the oncogenic signaling of FGFR2; therefore, it may be a therapeutic target for FGFR2-addicted DGC. This finding provides a strong rationale for the clinical evaluation of TfR1 inhibitors, such as monoclonal antibodies against TfR1 [97].

## 5. Perspective

RTK amplification is a key event in the malignant progression of DGC, and it is a potential therapeutic target. However, several obstacles hinder the clinical application of RTK inhibitors in the treatment of DGC. For example, despite obvious antitumor effects in preclinical models, several clinical trials have shown insufficient benefits of RTK inhibitors in the treatment of DGC [98]. This is most likely owing to the lack of biomarkers to select patients with tumors that are addicted to RTK signaling. In addition to assessing the amount of RTK proteins or genes, evaluating the activation status of downstream effectors and/or gene expression signatures may be necessary to select RTK-addicted tumors. Detailed tumor genome profiling may enable the accurate identification of RTK-addicted tumors and patients who can benefit from RTK-targeting therapies [64]. 

From a biological perspective, understanding the precise oncogenic RTK functions is important for the development of effective therapies. For instance, it remains unclear whether the specificity of downstream signaling exists among the different RTKs. FGFR2 and Met may have specific and overlapping roles, which may not be shared by other RTKs, such as HER2, in tumor malignancies, according to their embryogenic and morphogenic functions. This may confer biological advantages to DGC cells by reinforcing their invasive and metastatic phenotypes. Additionally, it remains unclear why RTK activation mechanisms differ between cancer types. In the case of Met, point mutations are the major activation mechanisms in renal cancer, whereas gene amplification is predominant in DGC. Similarly, Ras gene amplification is more prevalent in DGC, whereas point mutations are dominant in lung and pancreatic cancers. These facts raise the possibility that the signaling downstream of RTK is different between activation patterns and/or is cell-context dependent. It is also critical to understand whether oncogenic RTK signaling caused by gene amplification elicits activation of specific downstream pathways. If downstream signaling differs between oncogenic and physiological RTK activation, targeting only oncogenic signaling would be an ideal therapeutic approach with minimal adverse effects. Thus, further detailed biological studies are necessary to develop effective targeted therapies against DGC harboring RTK gene amplification.

## Figures and Tables

**Figure 1 cancers-14-03750-f001:**
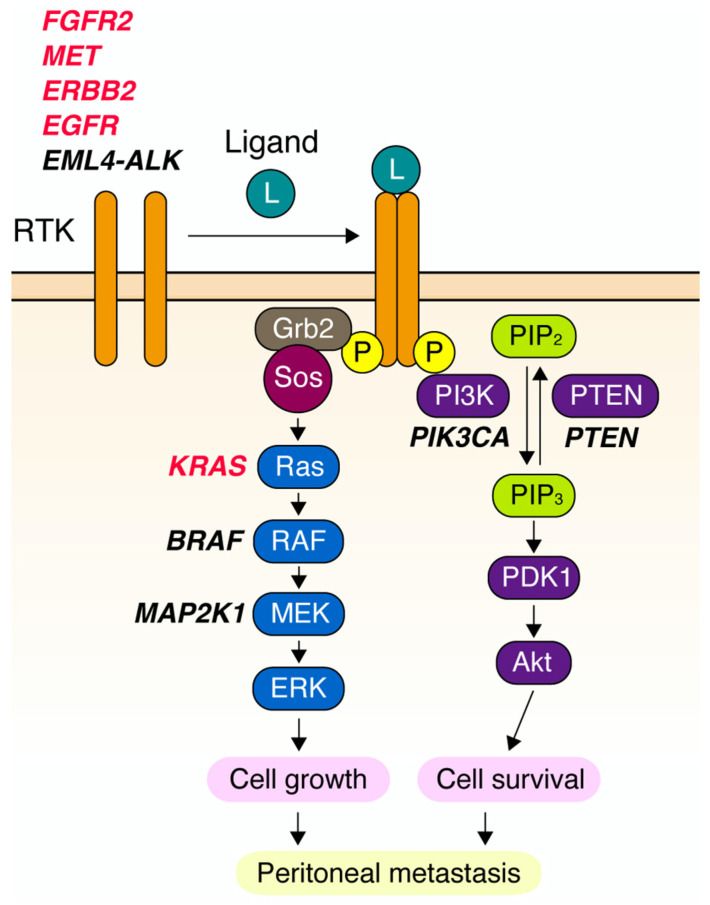
Receptor tyrosine kinase (RTK) signaling and gene alterations found in diffuse-type gastric carcinoma (DGC). Upon ligand binding, RTK molecules dimerize and transphosphorylate, which in turn, recruit a variety of intracellular signaling proteins. For example, Grb2/Sos bind to phosphorylated RTK and activate Ras signaling for cell growth. Analogously, phosphoinositide 3-kinase (PI3K) is recruited to activated RTK and generates phosphatidylinositol-3,4,5-triphosphate (PIP_3_), which is counteracted by PTEN, and activates Akt signaling for cell survival. Activation of these signaling pathways contributes to peritoneal metastasis. Genes encoding RTK signaling components altered in DGC are shown in bold and in oblique characters. Genes amplified in DGC are highlighted in red.

**Figure 2 cancers-14-03750-f002:**
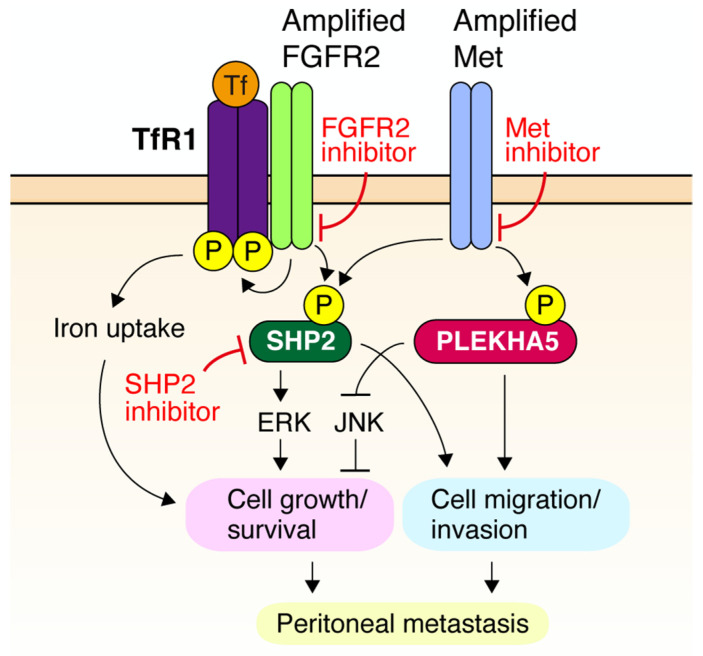
Cellular functions of PLEKHA5, SHP2, and Transferrin receptor 1 (TfR1 downstream of amplified RTKs in DGC. PLEKHA5 is tyrosine-phosphorylated downstream of amplified Met. Downregulation of PLEKHA5 induces apoptosis via JNK activation and blocks cell migration, invasion, and peritoneal metastasis in Met-addicted DGC cells. SHP2 is also tyrosine-phosphorylated downstream of amplified Met and FGFR. Inhibition of SHP2 blocks growth, migration, invasion, and peritoneal dissemination of Met-addicted DGC. TfR1 associates with FGFR2 and is tyrosine-phosphorylated. TfR1 promotes transferrin-mediated iron uptake, which is required for growth, survival, and peritoneal metastasis of FGFR2-addicted DGC cells.

**Table 1 cancers-14-03750-t001:** Gene amplification of RTKs in gastric cancer (GC).

Gene	Sample ^(1)^	Frequency (%) ^(2)^	Technique ^(3)^	Classification ^(4)^	Associated Phenotypes ^(5)^	Ref.
*EGFR*	Early GC Advanced GC Metastatic GC	0/20 (0%) 1/69 (1.4%) 3/32 (9.3%)	Southern blot		Metastatic tumor	[6]
*EGFR*	GC	6/70 (8.5%)	Slot blot	>2-fold	Large tumor, advanced stage, poor survival	[7]
*EGFR*	GEC	23/489 (4.7%)	FISH	EGFR/CEP7 > 2.2	Squamous cell carcinoma, poor survival	[8]
*EGFR*	GC	15/193 (7.7%)	SNP array	CNA		[9]
*EGFR*	GC	17/293 (5.8%)	SNP array	CNA		[10]
*EGFR*	GC	23/950 (2.4%)	FISH	EGFR/CEP7 ≥ 2		[11]
*EGFR*	GC ascites	4/98 (4.0%)	WGS	CNA > 5 × ploidy		[12]
*HER2*	Early GC Advanced GC Metastatic GC	0/20 (0%) 4/69 (5.7%) 8/32 (25%)	Southern blot		Metastatic tumor	[6]
*HER2*	GC	9/70 (12.8%)	Slot blot	>2-fold	Lymph node metastasis, poor survival	[7]
*HER2*	GC	15/128 (11.7%)	Southern blot	>2-fold	IGC, poor survival	[13]
*HER2*	GEC	45/489 (9.2%)	FISH	HER2/CEP17 > 2.2		[8]
*HER2*	GC	14/193 (7.2%)	SNP array	CNA	Poor survival	[9]
*HER2*	GC	38/293 (12.9%)	SNP array	CNA		[10]
*HER2*	Chinese GC	33/219 (15.0%)	FISH	HER2/CEP17 > 2		[14]
*HER2*	GC	51/475 (10.7%)	FISH	HER2/CEP17 > 2.2	Differentiated	[15]
*HER2*	Chinese GC Korean GC	30/204 (14.7%) 27/338 (7.9%)	FISH	HER2/CEP17 ≥ 2		[16]
*HER2*	GC	90/950 (9.4%)	FISH	HER2/CEP17 ≥ 2		[11]
*HER2*	GC/GEJC	756/3280 (23.0%)	FISH	HER2/CEP17 ≥ 2	IGC	[17]
*HER2*	Asian GC	32/134 (23.8%)	FISH	HER2/CEP17 ≥ 2	9/32 have Met coamplification	[18]
*HER2*	GC	33/208 (15.8%)	FISH/SISH	HER2/CEP17 ≥ 2	IGC, differentiated, heterogeneity is associated with DGC	[19]
*HER2*	GC/GEC	40/228 (17.5%)	FISH			[20]
*HER2*	GC ascites	5/98 (5.1%)	WGS	CNA > 5 × ploidy		[12]
*MET*	GC cell line Early GC Advanced GC SGC	6/11 (54.5%) 0/11 (0%) 15/64 (23.4%) 5/13 (38.4%)	Southern blot	≥3-fold		[21]
*MET*	GC	6/154 (3.8%)	FISH			[22]
*MET*	GC	7/70 (10%)	Slot blot	>2-fold	Infiltrative invasion, peritoneal dissemination, poor survival	[7]
*MET*	GC	13/128 (10.1%)	Southern blot	>2-fold	Lymph node metastasis, poor survival	[13]
*MET*	Stage II/III GC	21/216 (9.7%)	qPCR	≥5 copies	Poor survival	[23]
*MET*	Western GC	0/38 (0%)	FISH	MET/CEP7 > 2		[24]
*MET*	GC	100/472 (21.1%)	qPCR	>4 copies	Poor survival	[25]
*MET*	GEC	10/489 (2.0%)	FISH	MET/CEP7 > 2.2	High-grade, advanced stages, poor survival	[8]
*MET*	GC	8/193 (4.1%)	SNP array	CNA	Poor survival	[9]
*MET*	GC GC cell line	4/266 (1.5%) 3/11 (27.2%)	qPCR/FISH	≥4 copies		[26]
*MET*	GC	39/128 (30.4%)	qPCR	≥4 copies	Invasion, poor survival	[27]
*MET*	GC	12/293 (4.1%)	SNP array	CNA		[10]
*MET*	Chinese GC	12/196 (6.1%)	FISH	MET/CEP7 > 2	Lymph node and distant metastasis, Poor survival	[14]
*MET*	GC xenograft	5/30 (16.6%)	SNP array	CNA		[28]
*MET*	GC	12/950 (1.2%)	FISH	MET/CEP7 ≥ 2		[11]
*MET*	Chinese advanced or metastatic GC or GEJC	8/113 (7.0%)	FISH	MET/CEP7 > 2	DGC	[29]
*MET*	Asian GC	13/134 (9.7%)	FISH	MET/CEP7 ≥ 2	9/13 have HER2 coamplification	[18]
*MET*	GC	7/49 (14.2%)	CISH	MET/CEP7 ≥ 2		[30]
*MET*	GC ascites	7/98 (7.1%)	WGS	CNA > 5 × ploidy		[12]
*FGFR2*	GC GC xenograft	3/24 (12.5%) 2/13 (15.3%)	Southern blot			[31]
*FGFR2*	GC	3/154 (1.9%)	FISH			[22]
*FGFR2*	GC	18/193 (9.3%)	SNP array	CNA		[9]
*FGFR2*	GC	14/313 (4.4%)	FISH	FGFR2/CEP10 ≥ 2	Invasion, metastasis, poor survival	[32]
*FGFR2*	Chinese GC Chinese GC Caucasian GC	3/131 (2.2%) 9/197 (4.5%) 7/97 (7.2%)	aCGH FISH	log ratio > 0.8 FGFR2/CEP10 ≥ 2		[33]
*FGFR2*	GC	3/171 (1.7%)	FISH	FGFR2/CEP10 ≥ 2	Poor survival	[34]
*FGFR2*	GC	15/293 (5.1%)	SNP array	CNA		[10]
*FGFR2*	GC cell line GC	4/38 (10.5%) 24/482 (4.9%)	FISH qRT-PCR	FGFR2/CEP10 ≥ 2 > 4 copies		[35]
*FGFR2*	Chinese GC	10/198 (5.0%)	FISH	FGFR2/CEP10 > 2		[14]
*FGFR2*	UK GC Chinese GC Korean GC	30/408 (7.3%) 9/197 (4.4%) 15/356 (4.2%)	FISH	FGFR2/CEP10 ≥ 2	Lymph node metastasis and poor survival	[16]
*FGFR2*	GC	5/188 (2.6%)	FISH	FGFR2/CEP10 ≥ 2		[36]
*FGFR2*	GC	67/1974 (3.3%)	FISH	FGFR2/CEP10 > 2		[37]
*FGFR2*	GC (TCGA)	63/338 (18.6%)	WGS	CNA		[38]
*FGFR2*	GC ascites	11/98 (11.2%)	WGS	CNA > 5 × ploidy		[12]
*FGFR2*	Non-Asian GC	20/493 (4.0%)	CISH	FGFR2/CEP10 > 2		[39]

^(1)^ GC, gastric cancer; GEC, gastroesophageal cancer; GEJC, gastroesophageal junction cancer; SGC, scirrhous gastric cancer; TCGA, The Cancer Genome Atlas. ^(2)^ Numbers denote positive cases/total cases. ^(3)^ aCGH, array comparative genomic hybridization; CISH, chromogenic in situ hybridization; FISH, fluorescence in situ hybridization; qPCR, quantitative polymerase chain reaction; SISH, silver in situ hybridization; SNP, single nucleotide polymorphism; WGS, whole-genome sequencing. ^(4)^ CEP, chromosome enumerating probe; CNA, copy number alteration. ^(5)^ DGC, diffuse-type gastric cancer; IGC, intestinal-type gastric cancer.

**Table 2 cancers-14-03750-t002:** Gene amplification of *EGFR*, *ERBB2*, *MET*, and *FGFR2* in GC in publicly available datasets.

Gene	Dataset	Amplified/Total Tumors	Frequency
*EGFR*	ICGC_TCGA2020	10/68	14.7%
	MSKCC2017	6/100	6%
	OrigiMed2020	23/850	2.7%
	TCGA_PanCancerAtlas_STAD	23/438	5.2%
	MSK2021	16/320	5%
	TCGA2014	17/293	5.8%
*ERBB2*	ICGC_TCGA2020	15/68	22.0%
	MSKCC2017	18/100	18%
	OrigiMed2020	68/850	8%
	TCGA_PanCancerAtlas_STAD	58/438	13.2%
	MSK2021	37/320	11.5%
	TCGA2014	38/293	12.9%
*MET*	ICGC_TCGA2020	6/68	8.8%
	MSKCC2017	4/100	4%
	OrigiMed2020	19/850	2.2%
	TCGA_PanCancerAtlas_STAD	12/438	2.7%
	MSK2021	11/320	3.4%
	TCGA2014	12/293	4.0%
*FGFR2*	ICGC_TCGA2020	2/68	2.9%
	MSKCC2017	2/100	2%
	OrigiMed2020	46/850	5.4%
	TCGA_PanCancerAtlas_STAD	19/438	4.3%
	MSK2021	12/320	3.7%
	TCGA2014	15/293	5.1%

Gene-amplification analysis of the indicated stomach adenocarcinoma datasets was performed using the cBioPortal (https://www.cbioportal.org/, accessed on 18 May 2022).

**Table 3 cancers-14-03750-t003:** Drugs targeting RTK signaling that are effective in DGC harboring gene amplification of RTKs in vitro or in preclinical models.

Drug	Type ^(1)^	Target	Inhibited Functions and Phenotypes	Refs.
ABN401	SMI	Met	Cell growth, survival, tumor growth	[49]
AMG 337	SMI	Met	Cell growth, survival, tumor growth	[50,51]
Cabozantinib	SMI	Met/VEGFR2	Cell growth	[52]
Capmatinib/ INC280	SMI	Met	Cell growth, peritoneal metastasis	[12,53]
Crizotinib/PF-02341066	SMI	Met/ALK	Cell growth, survival, tumor growth	[54,55]
E7050	SMI	Met/VEGFR2	Cell growth, tumor growth, angiogenesis, peritoneal metastasis	[56]
Foretinib/GSK1363089	SMI	Met/VEGFR/PDGFRβ/Tie-2/RON/AXL	Cell growth	[57,58]
JNJ38877605	SMI	Met	Cell growth, survival	[26,55]
PHA-665752	SMI	Met	Cell growth, survival, tumor growth, peritoneal metastasis, ascites formation	[25,55,59,60]
S49076	SMI	Met/FGFR1-3/AXL	Cell growth, tumor growth	[61]
Savolitinib/Volitinib	SMI	Met	Cell growth, tumor growth	[30,62,63,64]
SGX523	SMI	Met	Cell growth, survival	[26]
SU11274	SMI	Met	Cell growth, survival, migration, peritoneal metastasis	[65]
Tivantinib/ARQ197	SMI	Met	Cell growth, survival	[58,66]
ABT-700	mAb	Met	Cell growth, survival, tumor growth	[18]
SAIT301	mAb	Met	Cell growth	[58]
Sym015	mAb	Met	Cell growth	[58]
P3D12-vc-MMAF	ADC	Met	Cell survival, tumor growth	[59]
AZD4547	SMI	FGFR1-3	Cell growth, tumor growth	[33,67]
Compound 23d	SMI	FGFR1-4	Cell growth, survival, tumor growth	[68]
Dovitinib	SMI	FGFR/VEGFR	Cell growth, survival, tumor growth	[9]
Erdafitinib/JNJ-42756493	SMI	FGFR1-4	Cell growth, tumor growth	[68,69]
Futibatinib	SMI	FGFR1-4	Cell growth, tumor growth	[70]
Infigratinib/BGJ398	SMI	FGFR1-3	Cell growth, peritoneal metastasis	[12,67]
Ki23057	SMI	FGFR1, 2/VEGFR/PDGFR/c-Kit	Cell growth, survival, tumor growth, peritoneal metastasis, lymph node metastasis, ascites formation	[71,72]
LY2874455	SMI	FGFR1-4	Tumor growth	[73]
Nintedanib	SMI	FGFR1-3/VEGFR1-3/PDGFRα, β	Cell growth	[74]
Pazopanib	SMI	FGFR/VEGFR/PDGFR/c-Kit	Cell growth, cell survival	[35]
PD173074	SMI	FGFR1-3	Cell growth, survival	[35,55,75]
Ponatinib/AP24534	SMI	FGFR/Bcr-Abl/VEGFR/PDGFR/Src	Cell growth, tumor growth	[76]
SOMCL-085	SMI	FGFR/VEGFR/PDGFR	Cell growth, tumor growth	[77]
DGY-09-192	PROTAC	FGFR1, 2	Cell growth	[78]
Bemarituzumab	mAb	FGFR2b	Cell growth, tumor growth	[79]
PRO-007	mAb	FGFR2	Cell growth, invasion	[80]
BAY 1187982	ADC	FGFR2	Tumor growth	[81]
Osimertinib	SMI	EGFR	Cell growth	[12]
SHP099	SMI	SHP2	Cell growth, migration, invasion, peritoneal metastasis, ascites formation	[82]
PI-103	SMI	PI3K	Tumor growth	[83]

^(1)^ SMI, small molecule inhibitor; mAb, monoclonal antibody; ADC, antibody–drug conjugate; PROTAC, proteolysis targeting chimera.

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
