# Peer review of "Receptor Tyrosine Kinases Amplified in Diffuse-Type Gastric Carcinoma: Potential Targeted Therapies and Novel Downstream Effectors"

_cancers, 2022, doi:10.3390/cancers14153750_

Round 1

Reviewer 1 Report

This present review article by Yamaguchi et al discusses the importance of oncogenic receptor tyrosine kinase (RTK) signalling in Diffuse-type gastric carcinoma (DGC) progression. This review correlates RTK amplification and involvement in peritoneal metastasis of DGC. Eventually, they explain the targeting strategies of DGC in Table 3. I am in principle supportive of accepting this work for publication. However, I have few suggestions to improve the review for publication.

Minor

In table 3, I suggest the author to include treatment response or outcome as a separate column. 

Author Response

Thank you very much for your constructive suggestion. However, most of the drugs listed in Table 3 are only tested in vitro or in preclinical models. Although some of the drugs are currently tested in clinical trials, conclusive results are not yet available. Moreover, the clinical trials are not restricted to DGC. Thus, we think that, at present, it is very difficult to include such clinical information in Table 3. Instead, we changed the title of Table 3 to clarify that the results are obtained in vitro or in preclinical models.

Reviewer 2 Report

Excellent and comprehensive work on the subject. Very important for general and practical knowledge with future implications in oncology

Author Response

Thank you very much for kindly reviewing our manuscript. I really appreciate your comment.

Reviewer 3 Report

This is a solid review of RTK and provides an emerging prospective on novel treatments for DGCs.

Through only a small amount of DGC are heriditary, CDH1/CTNNA1 mutations lead also to the DGC phenotype.  Could some comment be made on the connection between cadherin expression and/or the loss thereof and RTK signalling?

Author Response

Thank you very much for your comment. I agree with the reviewer that providing information on the relation between E-cadherin and RTK signaling in DGC is very important. Accordingly, in the revised manuscript, we added a paragraph describing E-cadherin and RTK signaling in lines 142-148.